# Optimal Design of a Fiber-Reinforced Plastic Composite Sandwich Structure for the Base Plate of Aircraft Pallets In Order to Reduce Weight

**DOI:** 10.3390/polym13050834

**Published:** 2021-03-09

**Authors:** Alaa Al-Fatlawi, Károly Jármai, György Kovács

**Affiliations:** 1Faculty of Mechanical Engineering and Informatics, University of Miskolc, Egyetemváros, H-3515 Miskolc, Hungary; vegyalaa@uni-miskolc.hu (A.A.-F.); jarmai@uni-miskolc.hu (K.J.); 2Faculty of Mechanical Engineering, University of Kufa, Al-Najaf 54001, Iraq

**Keywords:** application of fiber reinforced plastic composites, phenolic and epoxy woven glass fiber laminates, epoxy woven carbon fiber laminates, aircraft pallets, weight optimization method, case study, weight saving

## Abstract

The application of fiber-reinforced plastic (FRP) composite materials instead of metals, due to the low density of FRP materials, results in weight savings in the base plates of aircraft pallets. Lower weight leads to lower fuel consumption of the aircraft and thereby less environmental damage. The study aimed to investigate replacing the currently used aluminum base plates of aircraft pallets with composite sandwich plates to reduce the weight of the pallets, thereby the weight of the unit loads transported by aircraft. The newly constructed sandwich base plate consists of an aluminum honeycomb core and FRP composite face-sheets. First, we made experimental tests and numerical calculations for the investigated FRP sandwich panel to validate the applicability of the calculation method. Next, the mechanical properties of 40 different layer-combinations of 4 different FRP face-sheet materials (phenolic woven glass fiber; epoxy woven glass fiber; epoxy woven carbon fiber; and hybrid layers) were investigated using the Digimat-HC modeling program in order to find the appropriate face-sheet construction. Face-sheets were built up in 1, 2, 4, 6 or 8 layers with sets of fiber orientations including cross-ply (0°, 90°) and/or angle-ply (±45°). The weight optimization method was elaborated considering 9 design constraints: stiffness, deflection, skin stress, core shear stress, facing stress, overall buckling, shear crimping, skin wrinkling, and intracell buckling. A case study for the base plate of an aircraft pallet was introduced to validate the optimization procedure carried out using the Matlab (Interior Point Algorithm) and Excel Solver (Generalized Reduced Gradient Nonlinear Algorithm) programs. In the case study, the weight of the optimal structure (epoxy woven carbon fiber face-sheets) was 27 kg, which provides weight savings of 66% compared to the standard aluminum pallet. The article’s main added value is the elaboration and implementation of an optimization method that results in significant weight savings and thus lower fuel consumption of aircraft.

## 1. Introduction

The most commonly used type of composite is the fiber-reinforced plastic (FRP) composite, in which the materials consist of a basic matrix (e.g., resins) and a strengthening phase, i.e., fibers. FRP composite materials are much more advantageous than traditional metal materials (e.g., steel, aluminum) for many purposes. The required material content for a given application can be provided by selecting the suitable types, properties and proportions of different components.

The main significance of the research topic is that the application of FRP composite materials compared to metals results in a significant weight savings due to their low density. FRP composite materials’ further advantageous characteristics include high strength, high bending stiffness, corrosion resistance, good thermal insulation, and high vibration damping. FRP composite materials are applied in many industries (e.g., the construction, automotive, military, aerospace, and chemical industries) due to their above-mentioned characteristics. The application of composite materials in transport vehicles (air, water, and road) and loading units results in weight savings of vehicles and loading units. In the case of transport vehicles, this weight savings causes lower fuel consumption, thereby lower emissions and less environmental damage.

Our research aimed to replace the generally used aluminum base plates of aircraft pallets with the composite sandwich plates to reduce the weight of the pallets; thereby, the weight of the unit loads transported by aircraft will also be reduced. Therefore, a new light-weight composite sandwich base plate structure was constructed, which consists of an aluminum honeycomb core and FRP composite face-sheets. Four different FRP face-sheet materials were investigated: (1) phenolic woven glass fiber, (2) epoxy woven glass fiber, (3) epoxy woven carbon fiber and (4) hybrid (combination of epoxy woven glass fiber and epoxy woven carbon fiber) layers. The epoxy woven carbon fiber having higher stiffness to weight ratio compared to epoxy woven glass fiber. While the epoxy woven glass fiber having higher strength to weight ratio and more flexible compared to epoxy woven carbon fiber.

Epoxy resin is a polymer while phenolic resin is a synthetic polymer with versatile properties such as thermal stability, chemical resistance, fire resistance, and dimensional stability make it suitable for a wide range of applications. Phenolic and epoxy resins have been used in the composites industry as adhesives [1]. The epoxy resin has excellent mechanical performance, good environmental resistance, high toughness and easy processing. While, the phenolic resin has excellent fire resistance, good temperature resistance, low smoke and toxic emissions, rapid cure, and economic processing. FRP composite sandwich structures are geometrically more complex than monolithic constructions. Design and optimization methods of FRP composite sandwich structures are much more complex compared to homogenous monolithic structures [2,3,4]. 

The Digimat-HC program (version 2017.0, MSC Software, Irvine, CA, USA) is a multi-scale tool for the modeling of bending tests of honeycomb sandwich panels. It is a complete, simple, accurate, and flexible software tool dedicated to honeycomb sandwich structures. The Digimat-HC program takes into consideration the effect of the microstructure for both the core and the skins of the sandwich. For the honeycomb core, the homogenized properties are computed by Digimat-HC based on the geometry of the honeycomb unit cell. For the skins the same option is available. Skin is made of several layers piled up in a given order, with given orientations and thickness. Each layer can be defined at the macro or micro level.

The structure, main essences and added values of the article are the following: First, during our research, experimental tests and numerical calculations were carried out for the investigated FRP sandwich panels (phenolic woven glass fiber face-sheets and aluminum honeycomb core) to validate the applicability of the calculation methods (Section 3.1). Then the mechanical characteristics of 40 different layer-combinations of the above mentioned four FRP composite face-sheet materials were investigated using the Digimat-HC modeling program in order to find suitable face-sheet constructions with the FRP types (Section 3.2).

Next, a newly elaborated optimization method is introduced in the article. During the optimization the objective function was the weight objective function, because the most important design aim was the weight saving in case of our application. Furthermore, nine design constraints were taken into consideration during the optimization: total stiffness, total deflection, skin stress, core shear stress, skin facing stress, overall buckling, shear crimping, skin wrinkling, and intracell buckling. The optimization was carried out using the Matlab (Interior Point Algorithm) version R2018a, MathWorks, Inc., Natick, MA, USA and Excel Solver (Generalized Reduced Gradient Nonlinear Algorithm), Microsoft Excel 2010, Microsoft Corporation, Redmond, WA, USA programs (Section 3.3). 

Furthermore, a case study is described in order to confirm the practical applicability of the newly elaborated optimization method. In the case study, the base plate’s optimization procedure of a military aircraft pallet was introduced. In the case study, the optimal FRP type (which is the epoxy woven carbon fiber face-sheets) and construction of the base plates were determined instead of aluminum base plate (Section 4.3). Fuel cost saving and carbon saving caused by weight saving for the FRP composite sandwich base plate compared to the conventional aluminum base plate of aircraft pallets are introduced in Section 5.

The calculation method of composite face-sheets was solved using the Laminator program, that analysis laminated composite face-sheets according to classical lamination plate theory and the ply failure calculation based on Tsai–Hill failure criteria.

The main added value of the study is the elaboration and implementation of an optimization method for a base plate of an aircraft pallet which results in significant weight savings and less fuel consumption of aircrafts, thereby lower emissions and less environmental damage. The efficiency of the newly elaborated optimization method was confirmed by the case study.

## 2. FRP Composite Sandwich Structures

FRP composite sandwich structures have become common in engineering applications over the past 40 years. The application of FRP structures ranges from the aerospace and automobile industry to structural applications. Expanded FRP structure production reached an astonishing degree of automation in the first decade of the 21st century [2,3].

The composite sandwich structure provides low density and relative out-of-plane compression and shear properties. Honeycomb structures are natural or man-made structures that have the architecture of a honeycomb to reduce the amount of materials used in industrial applications to achieve minimum weight and cost of the structure [4].

There is interest in investigating these honeycomb structure’s performance and efficiency in multi-disciplinary application due to its high specific strength. Wang et al. studied the effects of aluminum honeycomb core thickness and density on the laminate material properties by three-point bending and panel peeling tests [5]. Yan et al. discussed the effects of face-sheet materials on the mechanical properties of aluminum foam sandwich under three-point bending by using electronic universal tensile testing machine [6]. Iyer et al. investigated a comparative study between three points and four points bending of sandwich composites made of rigid foam core and glass epoxy skin [7]. Inés and Almeida studied the structural behavior of FRP composite sandwich panels for applications in the construction industry [8]. 

Petras et al. investigated the flexural behavior of new generation FRP composite sandwich beams made up of glass fiber-reinforced polymer skins and modified phenolic core material by using 4-point static bending test to determine their strength and failure mechanisms in the flatwise and the edgewise positions [9]. Zhang studied an equivalent laminated model with three layers to simulate the behavior of the aluminum honeycomb sandwich panel with positive hexagon core [10]. Aborehab et al. discussed the mechanical behavior of an aluminum honeycomb structure exposed to flat-wise compressive and flexural testing. They proposed an equivalent finite element model based upon the sandwich theory to simulate the flexural testing’s elastic behavior and compare computational and experimental results [11]. 

Many studies have focused on how to obtain minimum weight and cost for honeycomb sandwich structures in some industrial applications [12,13,14,15,16,17]. Zaharia et al. performed compression, three-point bending and tensile tests to evaluate the performance of light-weight sandwich structures with different core topologies [18]. Yan et al. investigated the mechanical performance of the honeycomb sandwich structure with face-sheet/core debonding under a compressive load by experimental and numerical methods [19]. Baca Lopez and Ahmad estimated the best material sandwich structured arrangement design to enhance the mechanical properties [20]. 

Peliński and Smardzewski determined the effect of thickness and type of wood-based facings on stiffness, strength, ability to absorb, and dissipate the energy of sandwich beams with an auxetic core [21]. Yan et al. conducted a large experiment on three typical blade sandwich structures to simulate the natural lightning-induced arc effects [22]. Abada and Ibrahim investigated numerically the effectiveness of using ribbon shapes as an innovative core for sandwich structures subjected to blast loading [23]. Iftimiciuc et al. analyzed the structural performance of a novel pyramidal cellular core obtained through a mechanical expansion process [24]. Pereira and Fernandes employed an automated laminating line to manufacture sandwich panels for boards [25]. Mezeix and Wongtimnoi inspected bonding defects between the sandwich specimen’s multi-layers through nondestructive tests [26]. Galatas et al. fabricated process of low-density acrylonitrile butadiene styrenecarbon with carbon fiber reinforced polymer sandwich layers for unmanned aerial vehicle structure is proposed to improve the low mechanical strength and elastic modulus [27]. Pelanconi and Ortona reported on a nature-inspired, ultra-lightweight structure designed to optimize rigidity and density under bending loads [28]. Doluk et al. investigated the effect of layer orientations during milling and machining parameters for a sandwich structure composed of two materials, aluminum alloy and epoxy-carbon fibers [29,30,31]. Yuguo et al. proposed a finite element analysis method for grinding wheel and specimen of long fiber-reinforced ceramic matrix woven. This method was adopted to analyze the grinding process of a 2.5D woven quartz fiber-reinforced silicon dioxide ceramic matrix composite [32]. Soheil et al. presented a systematic approach toward localized failure inspection of internally pressurized laminated ellipsoidal woven composite domes. The domes were made of thin glass fiber reinforced polymer woven composite with (0,0,0), (0,30,0), (0,45,0), and (0,75,0) layups [33].

## 3. Materials and Methods

### 3.1. Experimental Investigation of FRP Composite Sandwich Specimens 

The FRP composite honeycomb sandwich construction is one of the most valued structural engineering innovations developed by the composites industry (Figure 1) [2]. The experimental tests included a four-point bending test in calculating the relationship between load *P* and displacement *δ_Exp_*. The specimens of sandwich panels are made of an aluminum honeycomb core and orthotropic composite materials face-sheets (Figure 2). The FRP composite face-sheets are made of phenolic woven glass fiber. Phenolic resin is a synthetic polymer. The fiber orientation of the composite face-sheets was cross-ply (0°, 90°). These specimens were made in the Kompozitor Ltd. Company (Budapest, Hungary). Numerical models are made for the same specimens using the Digimat-HC program to calculate the deflection, skin stress and core shear stress to compare with the experimental results as shown in Table 6 and Figures 7 and 8 (Section 4.1). The average skin stress and modulus can be determined with the following equations [34]:(1)$$\sigma =\frac{1}{8}\frac{Ps}{dbt_{f}}$$
(2)$$E=\frac{11}{384}\frac{P}{\delta }\frac{s^{3}}{bt_{f}d^{2}}$$
where $d=t_{c}+t_{f}$.

Here P is the total applied load, s is the specimen span, b is the panel width, d is the distance between face-sheet centers, $t_{f}$ is the face-sheet thickness,$\;t_{c}$ is the honeycomb core thickness, and δ is the deflection at mid-span. These equations are applicable for a symmetrical sandwich panel with thin face skins. These tests are achieved according to MIL-STD-401B Sec.5.2.4 [35].

The 4-point bending test provides four points of contact, two points of support and two points where loading is applied. The procedure of 4-point bending test is:Arrange the loading fixtures as shown in the appropriate Figure 2.Apply the load to the specimen through steel cylinders with loading pads.Measure the dimensions of the specimens and span length in mm.Apply the load at a constant rate that will cause the maximum load and record the maximum load.Load-deflection curves can be taken. A deflectometer can be used to measure the mid-span deflection.

### 3.2. Numerical Analysis of Different FRP Composite Sandwich Panels by Digimat-HC Program

The numerical models included a four-point bending test using Digimat-HC program. The technical data and configuration of the FRP composite sandwich are given as shown in Table 1 (see Figure 3) [36]. The numerical models of sandwich panels consist of an aluminum honeycomb core and different types of face-sheets, including composite materials. The composite face-sheets materials were one of the following: phenolic woven glass fiber, epoxy woven glass fiber, epoxy woven carbon fiber, or hybrid layers (a combination of epoxy woven glass fiber and epoxy woven carbon fiber). Every skin face-sheet is composed of 1, 2, 4, 6 or 8 layers. The fiber orientation in the face-sheets is cross-ply (0°, 90°) and/or angle-ply (±45°), where the mechanical properties of the core and face-sheets are shown in Table 2 and Table 3, respectively [36]. In this study, the mean vertical displacement at mid-section $\delta_{Num}$, equivalent stress in the skin face-sheets $\sigma_{skin}$ and equivalent shear stress in the honeycomb core $\tau_{core}$ were calculated. The numerical composite results consist of five main cases depending on face-sheets types of the sandwich panels. Every composite case study consists of sixteen different fiber orientations presented as shown in Tables 7–10 and Figures 9–11 (Section 4.2).

The analysis procedure of the Digimat-HC program is the following:Tab of Core

The core model is the base constituent of the sandwich structure, which is defined as the assembly of two skins (upper and lower face-sheets). The following parameters have to be defined: the name, the core model, and the core thickness. A honeycomb core can be defined at the micro and/or macro level. When it is defined at the micro-level, homogenized properties will be computed by the Digimat-HC based on the microstructure and its base material properties.

The tab of the microstructure is used to define information about the microstructure and the material of the honeycomb: the cell’s shape, the cell’s dimensions, and the base material properties. The tab of homogenized properties is used to define the homogenized macroscopic properties of the honeycomb. These properties can be either manually entered in the different fields or computed from the microstructure information (core/honeycomb) by a homogenization step (Figure A2, Figure A3 and Figure A4).

2.Tab of Layer

Defining layers is the second step of the analysis. The layer is the base constituent of the skins. The tab of homogenized properties is used to define the homogenized properties of the layer. The type of element is shell element. (Figure A5).

3.Tab of Sandwich

This tab (Pile-up definition) is used to define the composition (i.e., pile up sequence) of the skins. The first parameter that has to be defined is the number of layers. For each layer used in the pile-up, the following parameters have to be defined: the layer type, the layer orientation, and the layer thickness. (Figure A6 and Figure A7).

4.Tab of Loading

The type of loading is 4-point bending. The following parameters have to be defined: the panel geometry, the force (F), the width of the loading pads, the orientation of the core definition, the mesh refinement level is Fine mesh, and the symmetric boundary conditions not be used with a sandwich presenting skins that are ‘not equilibrated’. The finite element mesh will have around 9000 elements (see Figure A8).

5.Tab of Results

This tab is used to plot a 3D view of the sandwich. One case study of numerical analysis is explained in Appendix A (Figure A9).

### 3.3. Optimal Design of a Sandwich Base Plate Consisting of Aluminum Honeycomb Core and Fiber Reinforced Plastic Composite Face-Sheets—Case Study

This study aimed to investigate the replacement of the currently used aluminum base plate of aircraft pallets with a composite sandwich plate. The novel sandwich plate consists of an aluminum honeycomb core and FRP composite face-sheets. The investigated composite face-sheets consist of layers of a phenolic woven glass fiber, an epoxy woven glass fiber, an epoxy woven carbon fiber, or a hybrid composite (a combination of epoxy woven glass fiber and epoxy woven carbon fiber). Each face-sheet is composed of 2, 4, 6, or 8 layers. The layup of the fibers of the face-sheets was restricted to sets of plies having orientation angles of cross-ply (0°, 90°) and/or angle-ply (±45°). In all, 40 different layer combinations of 4 different FRP face-sheet materials are investigated, as discussed in Section 4.3. 

The pallet is a durable and robust freight pallet for efficient and cost-effective cargo transportation. This case study aimed to design a light-weight composite sandwich plate consisting of an aluminum honeycomb core with different types of face-sheets. The elaborated structural model could be used for manufacturing a base plate of aircraft cargo pallets to fulfill the requirements of military aircraft. The purpose of the application of light-weight pallet is to provide considerable savings in weight compared to the conventional aluminum sheet pallet, as shown in Figure 4 [37].

The pallets have dimensions of 3175 mm by 2235 mm and are supported by six frames (to distribute loads evenly over a larger area) which work in parallel inside the aircraft as shown in Figure 5 [2]. The dimensions of the rollers used to move the pallet into the airplane, as shown in Figure 6 [2]. The pallet design used today consists of a solid 4.2 mm thick Aluminum plate which weighs approximately 80 kg. The value of 1 kg of reduced weight is approximately EUR 163 per year. The total load on the pallet is 6800 kg, uniformly distributed.

Moreover, the pallet should be able to sustain an extra acceleration of 1.5 g, so the total load 2.5 g (1 g + 1.5 g extra acceleration). The maximum deformation may not exceed 50 mm. The loading system is approximated by studying the panels inscribed between the supports (with dimensions of 665 mm by 2235 mm).

The design parameters of the conventional base plate of the aircraft freight pallet (aluminum alloy Al7021-T6) are shown in Table 4. 

The plate’s boundary conditions are simply supported along the long edges and free along the shorter edges (see Table 5). The pallet is the centerpiece of the materials handling support system, and was designed in the late 1950s to provide more efficient intermodal cargo transfer for the air force. Today the pallet is a standard size platform for bundling and moving air cargo and serves as the primary air cargo pallet for military and many civilian cargo transport aircraft worldwide.

#### 3.3.1. Weight Optimization 

A methodology for weight optimization of an aircraft pallet’s base plate was elaborated because the primary design aim was the weight saving. Therefore, during the optimization the objective function was the weight of the sandwich plate. 

Furthermore, design constraints included stiffness, deflection, facing stress (bending load), core shear stress, skin stress (end loading), overall buckling, shear crimping, skin wrinkling, and intracell buckling. The design variables were core thickness and face-sheet thickness (number of FRP composite layers).

#### 3.3.2. Weight Objective Function

The total weight of the FRP composite sandwich structures, which including the weight of upper and lower face-sheets and honeycomb core (the weight of adhesive bond is neglected), was minimized using Matlab (Interior Point Algorithm) and Excel Solver (Generalized Reduced Gradient Nonlinear Algorithm) programs [38].

For the FRP composite sandwich structure, the equation of the total weight is:(3)$$W_{t}=W_{f}+W_{c}=2\;\rho_{f}lbt_{f}+\rho_{c}lbt_{c}$$
where $t_{f}=N_{l}t_{l}$.

$W_{t}$ is the total weight of the sandwich plate, $W_{f}$ and $W_{c}$ are the weight of face-sheets (upper and lower) and core, respectively, $\rho_{f}$ and $\rho_{c}$ are the density of the face-sheets and core, respectively, l and b are length and width of sandwich structure, respectively, $N_{l}$ is the number of laminates in the composite face-sheet, $t_{l}$ is the thickness of lamina, $t_{f}$ and $t_{c}$ are the thickness of the face-sheets and the core, respectively. 

#### 3.3.3. Design Variables

For an FRP composite sandwich structure in which the face-sheets are of composite materials, core thickness $t_{c}$ and the number of face-sheet layers $N_{l}$ were modified to achieve acceptable performance:(4)$$1\;\mathrm{mm}\;\leq \;t_{c,\;opt}\;\leq \;100\;\mathrm{mm}$$
(5)$$2\;\mathrm{layers}\;\leq \;N_{l,opt}\;\leq \;8\;\mathrm{layers}$$
where $t_{c,\;opt}$ and $N_{l,opt}$ are the optimum core thickness and the optimum number of face-sheet layers, respectively.

#### 3.3.4. Design Constraints

The design constraints of FRP composite sandwich structures include total stiffness (bending and shear stiffness), total deflection (bending and shear deflection), facing skin stress (bending load), core shear stress, facing skin stress (end loading), overall buckling (bending and shear critical buckling loads), shear crimping load, skin wrinkling (critical stress and load), and intracell buckling.

1.Total Stiffness (Bending Stiffness and Shear Stiffness)

The total stiffness constraint for the FRP composite sandwich structure includes the bending stiffness and shear stiffness:(6)$$D_{11,x}=D_{11}/\left(1-\nu_{12}^{f}\;\nu_{21}^{f}\right)\geq D_{min}=\frac{K_{b}pl^{4}}{\delta }$$
where $D_{11}=0.5d^{2}A_{11}^{f}+2D_{11}^{f}+2dB_{11}^{f}$
(7)$${\tilde{S}}_{11}=\frac{d^{2}}{t_{c}}\frac{E_{c}}{2\;\left(1+\nu_{c}\right)}$$
where$\;\nu_{12}^{f},\;\nu_{21}^{f}$ are the Poisson’s ratio of the face-sheet, $K_{b}$ is the bending deflection coefficient, p is the applied transverse and longitudinal load per unit area, l is the length of the sandwich structure, δ is the deflection of the sandwich structure, $A_{11}^{f},\;B_{11}^{f},\;D_{11}^{f}$ are the extensional, coupling and bending stiffness matrices of the face-sheets, respectively, ${\tilde{S}}_{11}$ is the shear stiffness of the composite sandwich structure, $t_{c}$ is the core thickness, d is the distance between facing skin centers, $t_{c}$ is the core thickness, $E_{c}$ is the modulus of elasticity of honeycomb core, and $\nu_{c}$ is the Poisson’s ratio of the honeycomb core [3]. The calculated bending stiffness of the sandwich structure in global coordinate $D_{11,x}$ must be higher than the minimum stiffness of the sandwich structure $D_{min}$, which was calculated by using the given data in Table 4 ($\delta =\delta_{max}$; $p=p_{max}$).

2.Total Deflection

The total deflection constraint of the composite face-sheet sandwich structure includes the bending deflection and shear deflection [39]:(8)$$\delta_{max}\geq \delta =\frac{K_{b}pl^{4}}{D_{11,x}}+\frac{K_{s}pl^{2}}{{\tilde{\;S}}_{11}}$$
where$\;K_{b}$ and $K_{s}$ are the bending deflection coefficient and shear deflection coefficient, respectively. The maximum deflection of the FRP composite sandwich structure $\delta_{max}$ has been given, which must be greater than the total deflection calculated δ.

3.Skin Stress

The constraint of the facing skin stress for the FRP composite sandwich structure is:(9)$$\sigma_{f,x}\geq \sigma_{f}=\frac{M}{dt_{f}b}$$
where M is the maximum bending moment. The typical yield strength of the composite material face-sheet in the x-direction $\sigma_{f,x},$ which has been calculated by using the Laminator program, must be greater than the calculated skin stress $\sigma_{f}$. 

4.Core Shear Stress

The core shear stress constraint of the FRP composite sandwich structure is:(10)$$\tau_{c,y}\geq \tau_{c}=\;\frac{F}{db}$$
where F is the maximum shear force. The typical shear stress in the transverse direction of the core material $\tau_{c,\;y}$, which has been given in Table 2, must be greater than the calculated core shear stress $\tau_{c}$. 

5.Skin Facing Stress (End Loading)

The skin facing stress constraint of the FRP composite sandwich structure is:(11)$$\sigma_{f,\;y}\geq \sigma_{f}=\;\frac{P}{2t_{f}b}$$

The typical yield strength of the composite face-sheet material in the y-direction $\sigma_{f,\;y}$ which has been calculated by using the Laminator program, must be greater than the calculated skin facing stress $\sigma_{f}$.

6.Overall Buckling (Bending Buckling and Shear Buckling)

The overall critical buckling loads of the FRP composite sandwich structure, which includes the bending buckling load and shear buckling load, is:(12)$$P_{b,\;cr}=\frac{\pi^{2}D_{11,x}}{\beta l^{2}+\frac{\pi^{2}D_{11,x}}{{\tilde{\;S}}_{11}}}\geq \frac{P}{b}$$
where the factor β depends on the boundary conditions and $P_{b,cr}$ is the overall critical buckling load. The calculated load at which overall critical buckling would occur is greater than the end load being applied per unit width. 

7.Shear Crimping

The shear crimping constraint of the FRP composite sandwich structure is:(13)$$P_{cr}=t_{c}G_{c}b\geq P$$
where $G_{c}=G_{w},$
$P_{cr}$ is the critical shear crimping load, $G_{W}$ is the core shear modulus in W-direction and $G_{c}$ is the core shear modulus given in Table 2. The calculated load at which shear crimping would occur is greater than the end load being applied P is given in Table 4.

8.Skin Wrinkling

The skin wrinkling constraint of the FRP composite sandwich structure is:(14)$$\sigma_{wr,\;cr}=0.5\;\sqrt[{3}]{E_{f,x}\;E_{c}\;G_{c}}\;\geq \sigma_{f,\;x}$$
where: $G_{c}=G_{L}$
(15)$$\sigma_{wr,\;cr}=0.5\;\sqrt[{3}]{E_{f,y\;}E_{c}\;G_{c}}\;\geq \sigma_{f,y}$$
where: $G_{c}=G_{W}$
(16)$$P_{wr,cr}=2\sqrt{D_{11}^{f}\frac{E_{c}}{\left(t_{c}/2\right)}}\geq \frac{P}{b}$$
where $E_{c}$ is the compression modulus of core and $G_{c}$. is the core shear modulus, given in Table 2, $E_{f,x}$ and $E_{f,y}$ are the Young’s modulus of elasticity of the composite face-sheet in the x-direction and y-direction, respectively, and $D_{11}^{f}$ is the element of laminate matrices. All of these parameters were calculated using the Laminator program.

The stress level at which skin wrinkling would occur $\sigma_{wr,\;cr}$ is well beyond the skin material typical yield strength in the x-direction $\sigma_{f,\;x}$ and in the y-direction $\sigma_{f,y}$ which was calculated with the Laminator program, so skin stress is more critical than skin wrinkling. The calculated load $P_{wr,cr}$ at which skin wrinkling would occur is greater than the end load per unit width being applied $\left(P/b\right)$. 

9.Intracell Buckling (Face-sheet Dimpling)

The face dimpling constraint of the FRP composite sandwich structure is:(17)$$\sigma_{f,\;cr}=\frac{2E_{f}}{\left(1-\nu_{12}^{f}\nu_{21}^{f}\right)}{\left[\frac{t_{f}}{s}\right]}^{2}\geq \sigma_{f,y}$$
where: $E_{f}=\sqrt{E_{f,x}E_{f,y}}$.

Here $E_{f}$, $E_{f,x}$ and $E_{f,y}$ are the average modulus of elasticity and the Young’s modulus of elasticity of composite face-sheet in the x-directions and y-directions, respectively, calculated using the Laminator program and s is the cell size given in Table 2.

The stress level at which intracell buckling would occur $\sigma_{f,\;cr}$ is well beyond the skin material typical yield strength $\sigma_{f,\;y}$ which has been calculated using the Laminator program, so skin stress is more critical than intracell buckling. 

## 4. Results and Discussion

### 4.1. Experimental Results of FRP Composite Sandwich Panels

Figure 7 and Figure 8 present the experimental results (four-point bending test), including the deflection-load curve for the honeycomb sandwich specimens, and the numerical results (four-point bending test) including deflection, skin stress, and core shear stress for the honeycomb sandwich models to the comparison. According to experimental and numerical results are shown in Table 6, the most efficient way to reduce the deflection of composite sandwich panels is to increase the honeycomb core thickness, thus increase the skin separation, and the most efficient way to reduce the skin stress and core shear stress is to increase the face-sheets thickness. Good agreement was found between experimental and numerical results. Only one figure of the experimental and numerical results is presented in this paper due to their similarity in behavior.

In Table 6, l is the specimen length, $\delta_{Exp}$ is the experimental deflection, $\delta_{Num}$ is the numerical deflection, $\sigma_{skin}$ is the numerical skin stress, $\tau_{core}$ is the numerical core shear stress and P/δ is the load-deflection curve slope. 

### 4.2. Numerical Results of Different FRP Composite Sandwich Panels

Table 7, Table 8, Table 9 and Table 10 show the effect of composite face-sheet thickness (*t_f_*) (number of layers and fiber orientations including cross-ply, angle-ply, and multidirectional ply) on the mean vertical displacement ($\delta_{Num}$), equivalent skin stress ($\sigma_{skin}$), and equivalent core shear stress ($\tau_{core}$) of FRP composite sandwich structure in case of 40 different layer-combinations of 4 different FRP face-sheet materials [(1) phenolic woven glass fiber, (2) epoxy woven glass fiber, (3) epoxy woven carbon fiber, and (4) hybrid layers]. 

Table 7 shows the numerical results of four-point bending test using the Digimat-HC program for sandwich panels consisting of an aluminum honeycomb core and FRP composite face-sheets of phenolic woven glass fiber.

Table 8 shows the numerical results of four-point bending test using the Digimat-HC program for sandwich panels consisting of an aluminum honeycomb core and FRP composite material face-sheets of epoxy woven glass fiber.

Table 9 shows the numerical results of four-point bending test using Digimat-HC program for sandwich panels consisting of an aluminum honeycomb core and FRP composite material face-sheets of epoxy woven carbon fiber.

Table 10 shows the numerical results of four-point bending test using Digimat-HC program for sandwich panels consisting of an aluminum honeycomb core and hybrid FRP composite material face-sheets (combination of epoxy woven carbon fiber and epoxy woven glass fiber).

#### Graphical Presentation of the Numerical Results of Different FRP Composite Sandwich Panels

Figure 9, Figure 10 and Figure 11 show the numerical results of different FRP composite sandwich panels in case of 40 different layer-combinations of 4 different FRP face-sheet materials (phenolic woven glass fiber, epoxy woven glass fiber, epoxy woven carbon fiber, and hybrid layers).

It can be concluded that the results relating to the phenolic glass fiber face-sheets (Table 7) and the epoxy woven glass fiber face-sheets (Table 8) are the same. Therefore, the characteristics of the sandwich panels consisting of both these types of face-sheets are presented by blue curves in the Figure 9, Figure 10 and Figure 11. The characteristics of the sandwich panels consisting of epoxy woven carbon fiber face-sheets are presented by red curves, while the characteristics of the sandwich panels consisting of hybrid composite face-sheets are presented by grey curves in Figure 9, Figure 10 and Figure 11.

Figure 9, Figure 10 and Figure 11 show the comparison of sandwich panels’ (1) mean vertical displacement ($\delta_{Num}$), (2) skin face-sheet stress ($\sigma_{skin}$), and (3) core shear stress ($\tau_{core}$) in case of the 4 different investigated FRP face-sheet materials and different fiber orientations. 

Figure 9 shows the comparison of mean vertical displacement (deflection) numerically using the Digimat-HC program (four-point bending test) for sandwich panels consisting of an aluminum honeycomb core ($t_{c}$ = 15 mm) and different composite material face-sheets of phenolic/epoxy woven glass fiber (because of the data of phenolic and epoxy woven glass fiber face-sheets are same; therefore, both types of face-sheets are presented by blue curve), epoxy woven carbon fiber and hybrid layer face-sheets with various numbers of layers $N_{l}$ and cross-ply (0°, 90°) and angle-ply (±45°) fiber orientation $\theta^{{}^{\circ}}$.

Figure 10 shows the comparison of equivalent skin face-sheet stress numerically using the Digimat-HC program (four-point bending test) for sandwich panels consisting of an aluminum honeycomb core and different composite material face-sheets of phenolic/epoxy woven glass fiber, epoxy woven carbon fiber and hybrid layers with various numbers of layers $N_{l}$ and cross-ply (0°, 90°) and angle-ply (±45°) fiber orientation $\theta^{{}^{\circ}}$.

Figure 11 shows the comparison of equivalent core shear stress numerically using the Digimat-HC program (four-point bending test) for sandwich panels consisting of an aluminum honeycomb core ($t_{c}$ = 15 mm) and different composite material face-sheets of phenolic/epoxy woven glass fiber, epoxy woven carbon fiber and hybrid layers with various numbers of layers $N_{l}$ and cross-ply (0°, 90°) fiber orientation $\theta^{{}^{\circ}}$.

It can be concluded that the mean vertical displacement, equivalent skin stress and equivalent core shear stress in case of epoxy woven carbon fiber face-sheets of the sandwich panels with fiber orientation cross-ply (0°, 90°) and angle-ply (±45°) are less than in case of the hybrid composite face-sheets, phenolic woven glass fiber and epoxy woven glass fiber face-sheets. While, the mean vertical displacement and equivalent core shear stress in case of cross ply (0°, 90°) fiber orientation face-sheets are less than angle ply (±45°) fiber orientation face-sheets of the sandwich panels. But, the equivalent skin stress in case of angle ply (±45°) fiber orientation is less than in case of cross ply (0°, 90°) fiber orientation face-sheets of the sandwich panels.

### 4.3. Optimization Results for a Base Plate of Aircraft Pallets

The final optimization results of military aircraft pallets include minimum total weight $W_{min,t}$ with optimum core thickness $t_{c,opt}$ and optimum face-sheet thickness $t_{f,opt}$ using the Excel Solver program and the Matlab program for the optimization. 

The optimization was achieved to minimize the weight of military aircraft pallets separately by applying the Excel Solver program (Generalized Reduced Gradient Nonlinear Algorithm) and the Matlab program (fmincon Solver Constrained Nonlinear Minimization/Interior Point Algorithm).

#### 4.3.1. Results of the Optimization by Applying the Excel Solver Program

The optimum results of weight optimization applying the Excel Solver program (Generalized Reduced Gradient Nonlinear Algorithm) for FRP composite material face-sheets and an aluminum honeycomb sandwich base plate for military aircraft pallets are shown in Table 11.

#### 4.3.2. Results of the Optimization by Applying the Matlab Program

The optimum results of weight optimization for FRP composite materials face-sheets, honeycomb sandwich base plate of military aircraft pallets obtained by applying the Matlab program (fmincon Solver Constrained Nonlinear Minimization/Interior Point Algorithm) are shown in Table 12.

The weight optimization was achieved to minimize the weight of military aircraft pallets separately by applying either the Excel Solver program (Table 11) or the Matlab program (Table 12). Based on the data of Table 11 and Table 12 it can be concluded that the results give good agreement between the two programs. 

Figure 12 shows the optimization results of the optimal FRP composite sandwich structure which provides the minimal weight.

It can be concluded that in the case study the primary design aim was the weight saving. The result of the weight optimization of the FRP composite sandwich structure for a base plate of aircraft pallet is 27.069 kg/piece, which provides the minimum weight. 

This optimal base plate consists of epoxy woven carbon fiber face-sheets consists of two layers with fiber orientation cross-ply (0°, 90°) (thicknesses are 0.6 mm) and aluminum honeycomb core (thickness is 24.27 mm). This optimal sandwich plate provides 66% weight saving compared to recently used aluminum base plate pallet (80 kg/piece).

## 5. Fuel Cost Saving and Carbon Saving Caused by Weight Saving 

According to the International Air Transport Association (IATA), every euro increase per barrel (42 gallons) drives an additional EUR 339 million in yearly fuel costs for passengers and cargo airlines. Fuel expenses now range from 25% to 40% of the total airline operating expenses. The new light-weight FRP composite freight pallets offer an enormous saving possibility compared to the conventional aluminum pallets. Data for calculating the fuel cost and discovering how much weight can be saved as well as carbon saving are shown in Table 13. Estimates from aircraft manufacturers and airlines vary greatly based on length of flight and type of aircraft but put operating costs at around 34 €/kg per year [40].

It can be concluded that the application of FRP composite materials instead of aluminum—due to the low density of FRP materials—would result in significant, 66% weight savings for the base plates of aircraft pallets (53 kg/pallet). The result of the saving in weight is proportional to saving in annual fuel cost or increases the payload of the aircraft. Lower weight causes lower fuel consumption of aircrafts, thereby less environmental damage. Due to the weight saving the fuel cost saving per year for one aircraft is 223,787 € and additional 27,943 € annual carbon cost can be saved. 

## 6. Factor of Safety (FoS) 

To designing an element or structure, the design engineers must consider many factors, such as safety factors. Safety is one of the most important qualities to be considered when creating parts or products. The term of “Factor of Safety” (FoS) or “Safety Factor” (SF) is most commonly. A basic equation to calculate FoS is to divide the ultimate (or maximum) stress by the typical (or working) stress, and the same for the load. Table 14 shows the factors of safety for optimum design constrains for the single base plate of military aircraft pallet.

## 7. Conclusions and Future Research

One of the most important advantages of the application of FRP composite materials compared to traditional metals is that their low density results in weight savings for base plates of aircraft pallets, which causes lower fuel consumption of aircrafts, thereby less environmental damage. Due to the above mentioned advantageous properties of FRP composites during our research the conventional aluminum base plates of aircraft pallets were replaced with FRP composite sandwich plates in order to reduce the weight of the pallets, thereby the weight of the unit loads transported by aircraft.

A new lightweight FRP composite sandwich base plate for an aircraft pallet structure was constructed which consists of aluminum honeycomb core and FRP composite face-sheets. During the construction of the face-sheets four different FRP materials were investigated: (1) phenolic woven glass fiber, (2) epoxy woven glass fiber, (3) epoxy woven carbon fiber and (4) hybrid (combination of epoxy woven glass fibers and epoxy woven carbon fibers) layers. Furthermore, the possible layer-combinations of FRP composite face-sheets were investigated. 

The mechanical properties of 40 different layer-combinations of the four different FRP face-sheet materials were calculated using the Digimat-HC modeling program in order to find the adequate face-sheet material and construction. Face-sheets were built up 1, 2, 4, 6 or 8 layers with sets of fiber orientations including cross-ply (0°, 90°) and/or angle-ply (±45°). The laminated composite panels were symmetric concerning the mid-plane of the sandwich panels.

Weight optimization methods were elaborated for the newly constructed light-weight FRP composite structure, because the most important design aim was the weight saving. During the optimization nine design constraints were taken into consideration: total stiffness; total deflection; skin stress; core shear stress; skin facing stress; overall buckling; shear crimping; skin wrinkling; intracell buckling. The optimization was carried out using both the Matlab (Interior Point Algorithm) and Excel Solver (Generalized Reduced Gradient Nonlinear Algorithm) programs. Good agreement was found between Excel Solver and Matlab results. One case study for the analysis of laminated composite plates based on classical laminated plate theory is explained in Appendix B.

A case study was carried out in order to confirm the practical applicability of the newly elaborated optimization method. In the case study, the optimization procedure of a base plate of a military aircraft pallet was introduced; furthermore, the optimal FRP type and construction of the pallet’s base plate was defined and compared with data for the conventional aluminum base plate.

The weight optimization of the FRP composite sandwich structure for a base plate of a military aircraft pallet yielded the minimum weight of 27.069 kg/piece. This optimal base plate consists of epoxy woven carbon fiber face-sheets consists of two layers with fiber orientation cross-ply (0°, 90°) (0.6 mm thick) and aluminum honeycomb core (24.27 mm thick). It can be concluded that in the case study the weight of the optimal lightweight FRP structure was 27 kg, which provides a 66% weight saving (53 kg) compared to the recently used conventional aluminum pallet (80 kg). 

The result of the saving in weight is proportional to saving in annual fuel cost. Lower weight causes lower fuel consumption of aircrafts. 

The study’s main added value is the elaboration and implementation of an optimization method for a base plate of aircraft pallets, which results in significant weight savings, thereby less fuel consumption of aircrafts and less environmental damage. The efficiency of the newly elaborated optimization method was confirmed by the case study.

It can be concluded relating to the future research that both the selection of the adequate material types and design of the appropriate structure for a given new application are very important. Furthermore, it can be summarized that the newly designed light-weight FRP composite sandwich structure is suggested in those applications where the primary aim is weight saving. 

In future research, the newly designed FRP sandwich structure and the newly elaborated optimization method can be used in other engineering applications, e.g., structural components of transport vehicles (air, water, road, rail) and elements of unit load devices. Furthermore, additional types of FRP composites and design constraints can be taken into consideration during the design and optimization procedures.

## Figures and Tables

**Figure 1 polymers-13-00834-f001:**
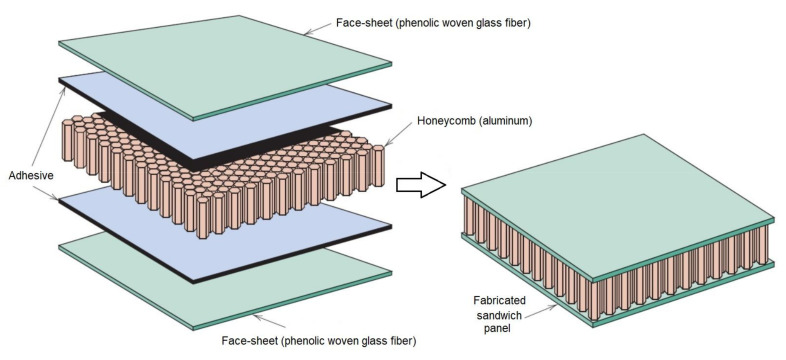
Construction of fiber-reinforced plastic (FRP) composite sandwich structure.

**Figure 2 polymers-13-00834-f002:**
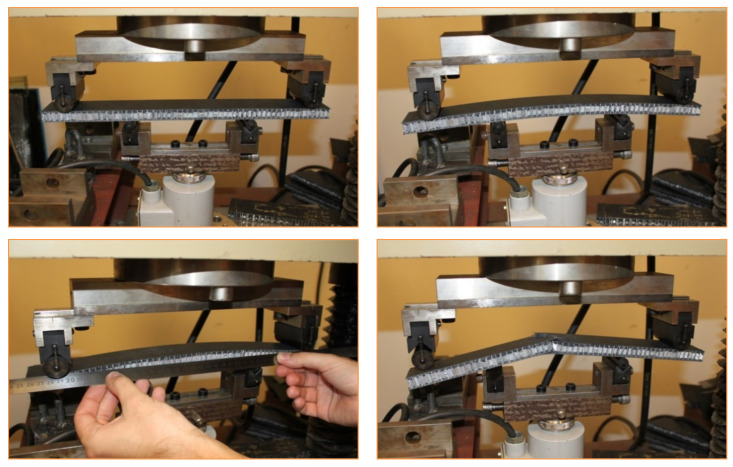
Experimental specimens (four-point bending test) for sandwich panels consisting of aluminum honeycomb core and phenolic woven glass fiber face-sheets.

**Figure 3 polymers-13-00834-f003:**
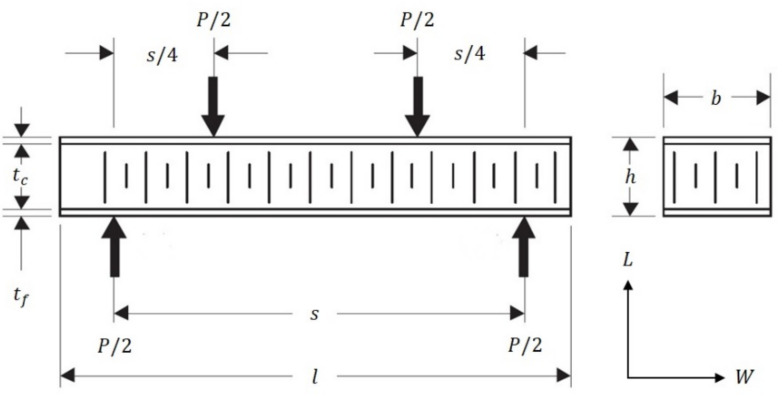
Set up and configuration of the FRP composite sandwich structure for a four-point bending test by applying the Digimat-HC program.

**Figure 4 polymers-13-00834-f004:**
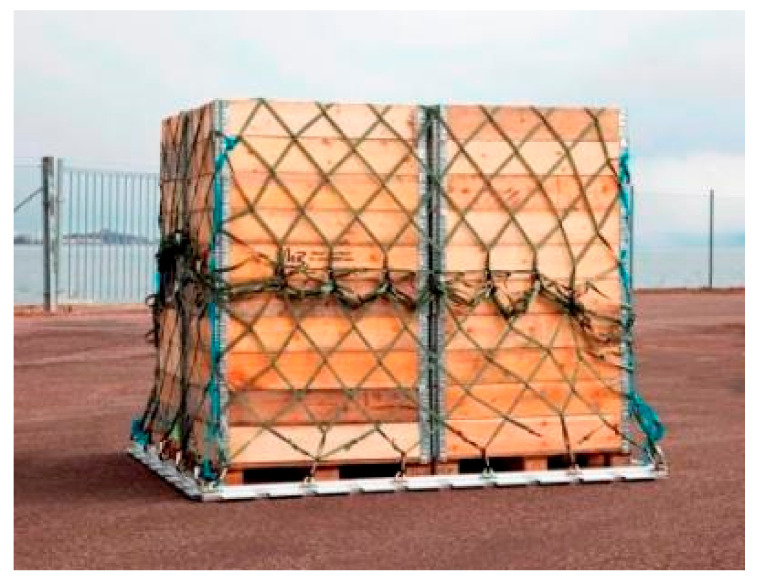
Base plate of a conventional aluminum sheet aircraft pallet.

**Figure 5 polymers-13-00834-f005:**
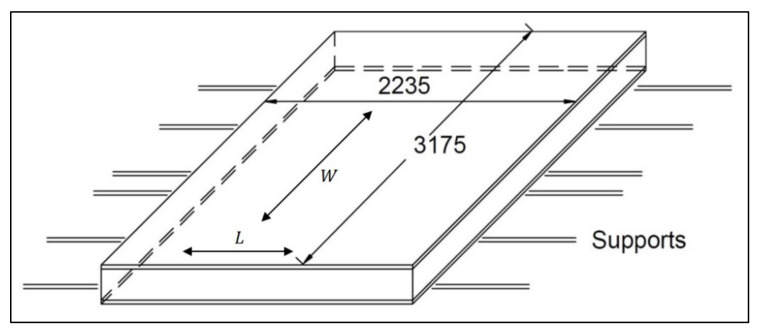
Dimensions of the base plate of military aircraft pallet with the supported beam, in [mm].

**Figure 6 polymers-13-00834-f006:**
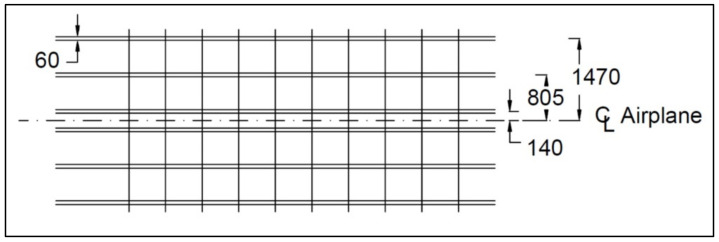
The dimensions of the rollers used to move the pallet into the airplane, in [mm].

**Figure 7 polymers-13-00834-f007:**
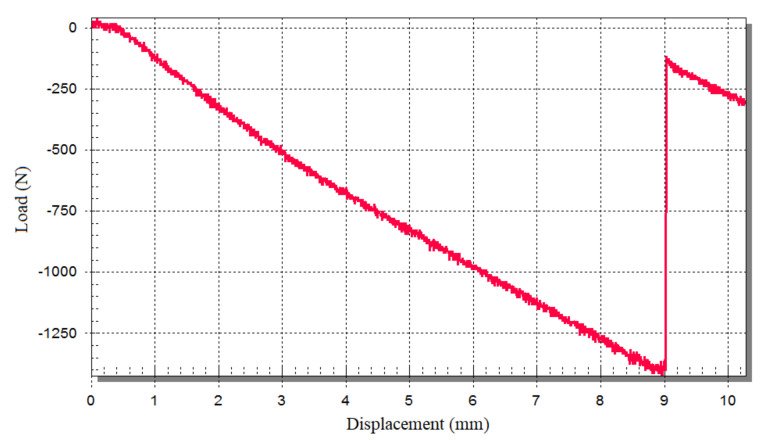
Experimental result (four-point bending test) for a specimen of the sandwich panel consisting of an aluminum honeycomb core ($t_{c}$ = 15 mm) and phenolic woven glass fiber face-sheets ($t_{f}$ = 1 mm).

**Figure 8 polymers-13-00834-f008:**
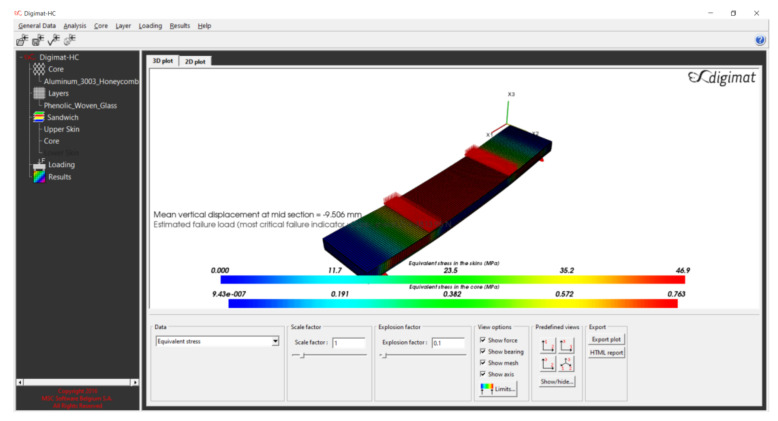
Numerical result (four-point bending test) for a specimen of the sandwich panel consisting of an aluminum honeycomb core ($t_{c}$ = 15 mm) and phenolic woven glass face-sheets ($t_{f}$ = 1 mm).

**Figure 9 polymers-13-00834-f009:**
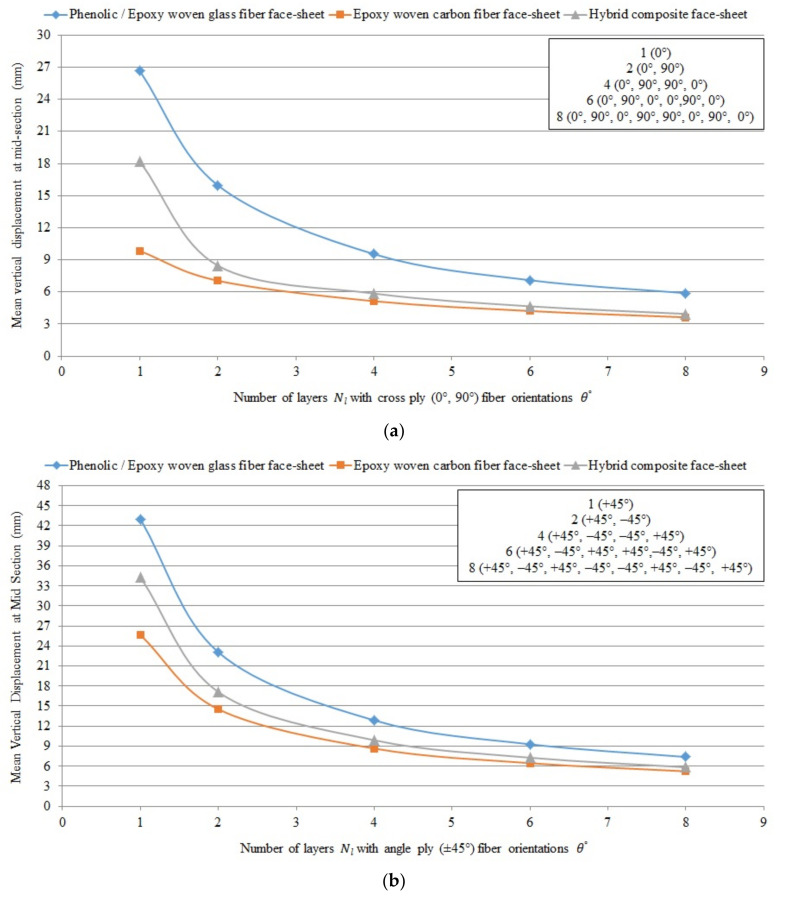
Comparison of deflection numerically for sandwich panels consisting of an aluminum honeycomb core and different composite material face-sheets of phenolic/epoxy woven glass fiber (because of the data of phenolic and epoxy woven glass fiber face-sheets are same; therefore, both types of face-sheets are presented by blue curve), epoxy woven carbon fiber and hybrid layer face-sheets with various numbers of layers $N_{l}$ and (**a**) cross-ply (0°, 90°) and (**b**) angle-ply (±45°) fiber orientation $\theta^{{}^{\circ}}$.

**Figure 10 polymers-13-00834-f010:**
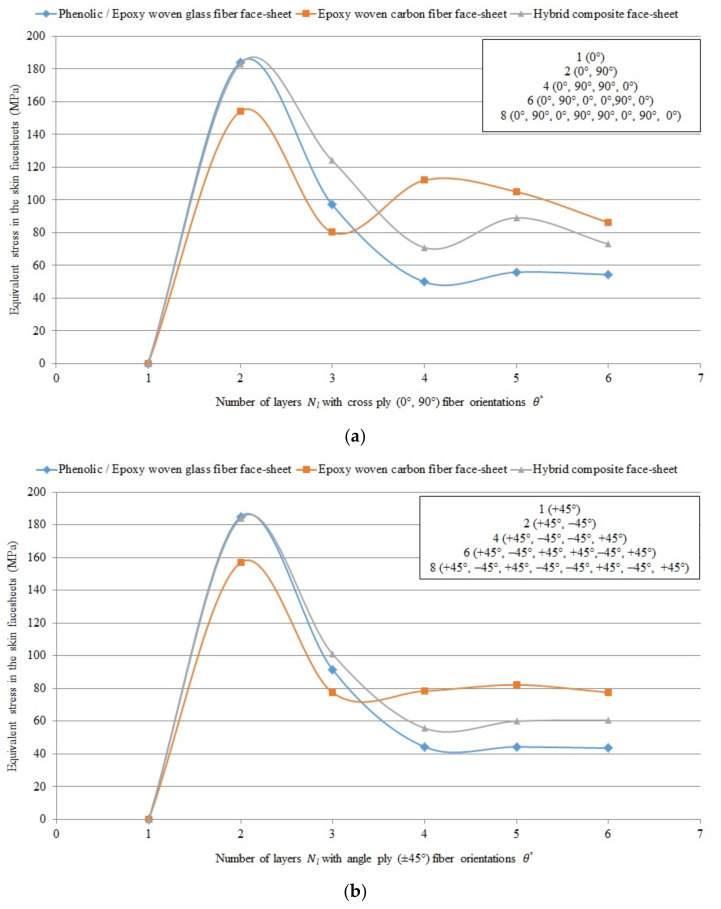
Comparison of skin face-sheet stress numerically for sandwich panels consisting of an aluminum honeycomb core and different composite material face-sheets of phenolic/epoxy woven glass fiber, epoxy woven carbon fiber and hybrid layers with various numbers of layers $N_{l}$ and (**a**) cross-ply (0°, 90°) and (**b**) angle-ply (±45°) fiber orientation $\theta^{{}^{\circ}}$.

**Figure 11 polymers-13-00834-f011:**
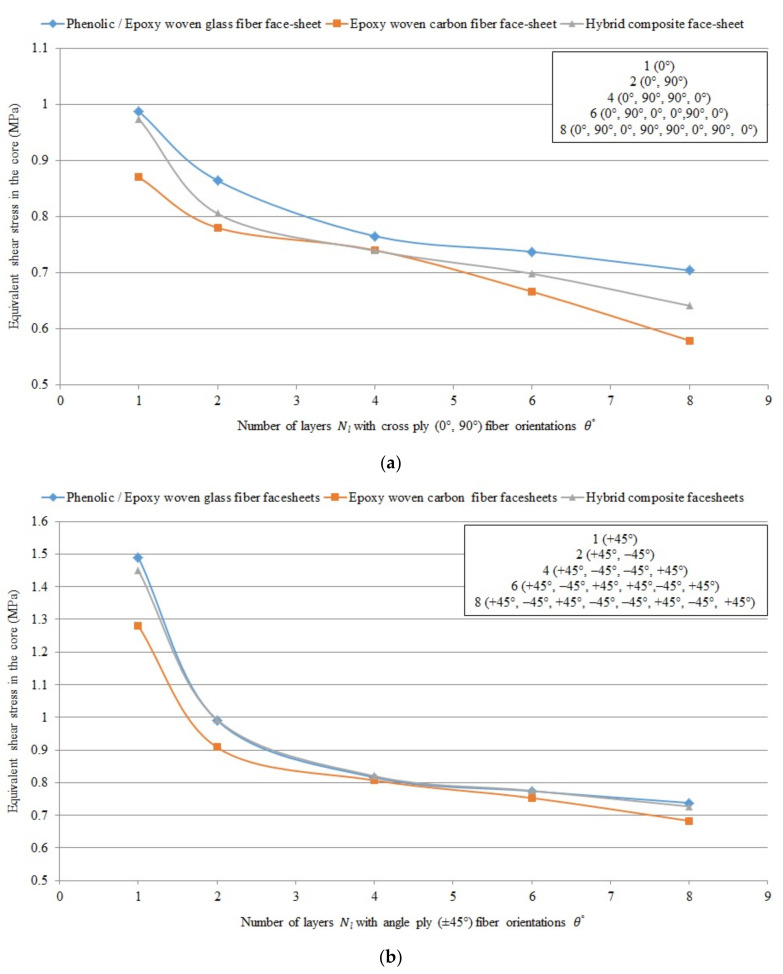
Comparison of core shear stress numerically for sandwich panels consisting of an aluminum honeycomb core and different composite material face-sheets of phenolic/epoxy woven glass fiber, epoxy woven carbon fiber and hybrid layers with various numbers of layers $N_{l}$ and (**a**) cross-ply (0°, 90°) and (**b**) angle-ply (±45°) fiber orientation $\theta^{{}^{\circ}}$.

**Figure 12 polymers-13-00834-f012:**
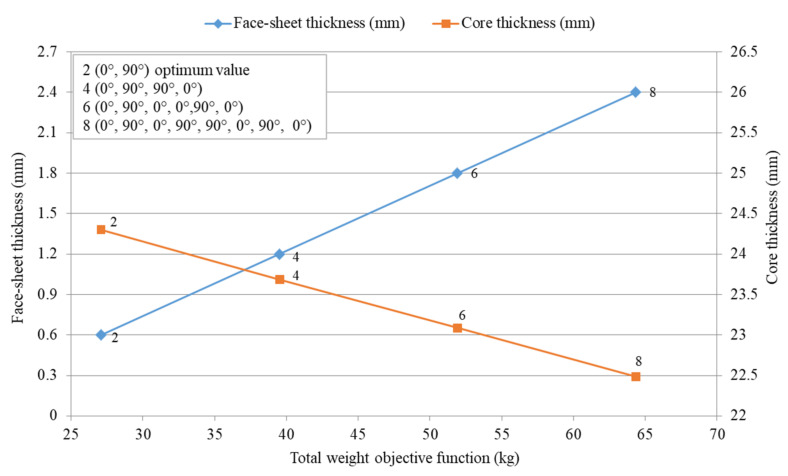
Minimum total weight objective function versus optimum face-sheet and core thicknesses for sandwich base plate of military aircraft pallets consisting of an aluminum honeycomb core and epoxy woven carbon fiber face-sheets with different number of layer $N_{l}$ and cross-ply (0°, 90°) fiber orientation $\theta^{{}^{\circ}}$.

**Table 1 polymers-13-00834-t001:** Technical data of fiber-reinforced plastic (FRP) composite sandwich models for Digimat-HC program.

Index	Length	Span	Width	Core Thickness	Face-Sheet Thickness	Load
	l	s	b	$t_{c}\;$	$t_{f}\;$	P
	[mm]	[mm]	[mm]	[mm]	[mm]	[N]
1	460	400	100	15	1	1400

**Table 2 polymers-13-00834-t002:** Engineering properties of aluminum honeycomb core materials.

Product Construction		Compression		Plate Shear			
Density	Cell Size	Stabilized		L-Direction		W-Direction	
		Strength	Modulus	Strength	Modulus	Strength	Modulus
[kg/m^3^]	[mm]	[MPa]	[MPa]	[MPa]	[MPa]	[MPa]	[MPa]
83	6	4.6	1000	2.4	440	1.5	220

**Table 3 polymers-13-00834-t003:** Engineering properties of the skin facing materials for FRP composite sandwich structure construction.

Facing Material	Typical Strength Tension/Compression [MPa]	Modulus of Elasticity Tension/Compression [GPa]	Poisson’s Ratio [μ]	Typical Cured Ply Thickness [mm]	Typical Weight per Ply [kg/m^2^]
Phenolic Woven Glass Fiber	400/360	20/17	0.13	0.25	0.47
Epoxy Woven Glass Fiber	600/550	20/17	0.13	0.25	0.47
Epoxy Woven Carbon Fiber	800/700	70/60	0.05	0.3	0.45

**Table 4 polymers-13-00834-t004:** Technical data for the conventional military pallet, aluminum alloy-Al7021-T6 [2,3].

Length	Width	Thickness	Deflection	Payload			Weight
l	b	t	$\delta_{max}\;$	$W_{max}\;$	P	p	$W_{t}\;$
[mm]	[mm]	[mm]	[mm]	[kg]	[N]	[Pa]	[kg]
3175	2235	4.2	50	6800	166,770	23,501.56	80

**Table 5 polymers-13-00834-t005:** Boundary conditions for the FRP composite sandwich structure [34,36].

Bending Deflection Coefficient	Shear Deflection Coefficient	Maximum Bending Moment	Maximum Shear Force	Buckling Factor
$K_{b}\;$	$K_{s}\;$	M	F	β
$\frac{5}{384}$	$\frac{1}{8}$	$\frac{Pl}{8}$	$\frac{P}{2}$	1

**Table 6 polymers-13-00834-t006:** Technical data and results of experimental tests by applying four-point bending test in the Kompozitor Ltd. Company and numerical models using the Digimat-HC program for FRP composite sandwich specimens set.

Index	Length	Span	Width	Core Thickness	Face-Sheet Thickness	Load	Stress	Shear	Deflection		Difference
	l	s	b	$t_{c}\;$	$t_{f}\;$	P	$\sigma_{skin}\;$	$\tau_{core}\;$	$\delta_{Exp}\;$	$\delta_{Num}\;$	
	[mm]	[mm]	[mm]	[mm]	[mm]	[N]	[MPa]	[MPa]	[mm]	[mm]	[%]
1	460	400	100	15	1	1400	46.9	0.76	9	9.5	5.62
2					1	1500	50.3	0.82	10.2	10.18	0.15
3					1	1600	53.6	0.87	11	10.86	1.24
4				19	2	1650	44.8	0.67	5.7	5.34	6.23
5					2	2000	54.4	0.82	6.5	6.48	0.32
6					2.5	1800	52.4	0.68	4.5	4.85	7.87
7					2.5	1900	50.5	0.74	5	5.36	7.14

**Table 7 polymers-13-00834-t007:** Numerical results for sandwich panels consisting of an aluminum honeycomb core (*t_c_* = 15 mm) and composite material face-sheets of phenolic woven glass fiber.

Type	$(1)\;\mathrm{Phenolic}\;\mathrm{Woven}\;\mathrm{Glass}\;\mathrm{Fiber}\;(t_{c}\;=\;15\;\mathrm{mm})\;$	$t_{f}\;$	$\delta_{Num}\;$	$\sigma_{skin}\;$	$\tau_{core}\;$
No.	$\mathrm{Number}\;\mathrm{of}\;\mathrm{Layers}\;N_{l}$ $\;\mathrm{and}\;\mathrm{Fiber}\;\mathrm{Orientations}\;\theta^{{}^{\circ}}$	[mm]	[mm]	[MPa]	[MPa]
1	1 (0°)	0.25	26.67	184	0.99
2	2 (0°, 90°)	0.5	15.98	97.1	0.86
3	4 (0°, 90°, 90°, 0°)	1	9.55	50	0.76
4	6 (0°, 90°, 0°, 0°,90°, 0°)	1.5	7.11	55.9	0.74
5	8 (0°, 90°, 0°, 90°, 90°, 0°, 90°, 0°)	2	5.89	54.4	0.7
6	1 (+45°)	0.25	42.98	185	1.49
7	2 (+45°, −45°)	0.5	23.06	91.5	0.99
8	4 (+45°, −45°, −45°, +45°)	1	12.87	44.4	0.83
9	6 (+45°, −45°, +45°, +45°, −45°, +45°)	1.5	9.29	44.4	0.77
10	8 (+45°, −45°, +45°, −45°, −45°, +45°, −45°, +45°)	2	7.38	43.6	0.74

**Table 8 polymers-13-00834-t008:** Numerical results for sandwich panels consisting of an aluminum honeycomb core (*t_c_* = 15 mm) and composite material face-sheets of epoxy woven glass fiber.

Type	$\left(2)\;\mathrm{Epoxy}\;\mathrm{Woven}\;\mathrm{Glass}\;\mathrm{Fiber}\;(t_{c}\;=\;15\;\mathrm{mm}\right)$	$t_{f}\;$	$\delta_{Num}\;$	$\sigma_{skin}\;$	$\tau_{core}\;$
No.	$\mathrm{Number}\;\mathrm{of}\;\mathrm{Layers}\;N_{l}$ $\;\mathrm{and}\;\mathrm{Fiber}\;\mathrm{Orientations}\;\theta^{{}^{\circ}}$	[mm]	[mm]	[MPa]	[MPa]
1	1 (0°)	0.25	26.67	184	0.99
2	2 (0°, 90°)	0.5	15.98	97.1	0.86
3	4 (0°, 90°, 90°, 0°)	1	9.55	50	0.76
4	6 (0°, 90°, 0°, 0°,90°, 0°)	1.5	7.11	55.9	0.74
5	8 (0°, 90°, 0°, 90°, 90°, 0°, 90°, 0°)	2	5.89	54.4	0.7
6	1 (+45°)	0.25	42.98	185	1.49
7	2 (+45°, −45°)	0.5	23.06	91.5	0.99
8	4 (+45°, −45°, −45°, +45°)	1	12.87	44.4	0.82
9	6 (+45°, −45°, +45°, +45°, −45°, +45°)	1.5	9.29	44.4	0.77
10	8 (+45°, −45°, +45°, −45°, −45°, +45°, −45°, +45°)	2	7.38	43.6	0.74

**Table 9 polymers-13-00834-t009:** Numerical results for sandwich panels consisting of an aluminum honeycomb core (*t_c_* = 15 mm) and composite material face-sheets of epoxy woven carbon fiber.

Type	$\left(3)\;\mathrm{Epoxy}\;\mathrm{Woven}\;\mathrm{Carbon}\;\mathrm{Fiber}\;(t_{c}\;=\;15\;\mathrm{mm}\right)$	$t_{f}\;$	$\delta_{Num}\;$	$\sigma_{skin}\;$	$\tau_{core}\;$
No.	$\mathrm{Number}\;\mathrm{of}\;\mathrm{Layers}\;N_{l}$ $\;\mathrm{and}\;\mathrm{Fiber}\;\mathrm{Orientations}\;\theta^{{}^{\circ}}$	[mm]	[mm]	[MPa]	[MPa]
1	1 (0°)	0.3	9.84	154	0.87
2	2 (0°, 90°)	0.6	7.06	80.2	0.78
3	4 (0°, 90°, 90°, 0°)	1.2	5.15	112	0.74
4	6 (0°, 90°, 0°, 0°,90°, 0°)	1.8	4.23	105	0.67
5	8 (0°, 90°, 0°, 90°, 90°, 0°, 90°, 0°)	2.4	3.64	86.2	0.58
6	1 (+45°)	0.3	25.66	157	1.28
7	2 (+45°, −45°)	0.6	14.53	77.5	0.91
8	4 (+45°, −45°, −45°, +45°)	1.2	8.65	78.5	0.81
9	6 (+45°, −45°, +45°, +45°, −45°, +45°)	1.8	6.46	82.3	0.75
10	8 (+45°, −45°, +45°, −45°, −45°, +45°, −45°, +45°)	2.4	5.23	77.6	0.68

**Table 10 polymers-13-00834-t010:** Numerical results for sandwich panels consisting of an aluminum honeycomb core ($t_{c}$ = 15 mm) and hybrid composite material face-sheets.

Type	$\left(4)\;\mathrm{Hybrid}\;\mathrm{Composite}\;\mathrm{Face}-\mathrm{Sheet}\;(t_{c}\;=\;15\;\mathrm{mm}\right)$	$t_{f}\;$	$\delta_{Num}\;$	$\sigma_{skin}\;$	$\tau_{core}\;$
No.	$\mathrm{Number}\;\mathrm{of}\;\mathrm{Layers}\;N_{l}$ $\;\mathrm{and}\;\mathrm{Fiber}\;\mathrm{Orientations}\;\theta^{{}^{\circ}}$	[mm]	[mm]	[MPa]	[MPa]
1	1 (0°)	0.3, 0.25	18.22	183	0.97
2	2 (0°, 90°)	0.55	8.47	124	0.8
3	4 (0°, 90°, 90°, 0°)	1.1	5.87	70.9	0.74
4	6 (0°, 90°, 0°, 0°,90°, 0°)	1.65	4.67	89.1	0.69
5	8 (0°, 90°, 0°, 90°, 90°, 0°, 90°, 0°)	2.2	3.959	73	0.64
6	1 (+45°)	0.3, 0.25	34.28	184	1.45
7	2 (+45°, −45°)	0.55	17.1	101	0.99
8	4 (+45°, −45°, −45°, +45°)	1.1	9.89	55.8	0.82
9	6 (+45°, −45°, +45°, +45°, −45°, +45°)	1.65	7.28	60.1	0.77
10	8 (+45°, −45°, +45°, −45°, −45°, +45°, −45°, +45°)	2.2	5.84	60.7	0.73

**Table 11 polymers-13-00834-t011:** Minimum weight objective function with optimum face-sheet thickness and optimum core thickness for the sandwich base plate consisting of an aluminum honeycomb core and different orthotropic FRP composite face-sheets [(1) Phenolic woven glass fiber, (2) Epoxy woven glass fiber, (3) Epoxy woven carbon fiber and (4) Hybrid layers] with different number of layer $N_{l}$ and fiber orientation $\theta^{{}^{\circ}}$.

Type	Number of Layers *N_l_*	Fiber Orientations θ°	Wmin,t [kg]	tf,opt [mm]	tc,opt [mm]
	Optimum Value				
Phenolic woven glass fiber face-sheet	4	(0°, 90°, 90°, 0°)	40.742	1	23.872
Epoxy woven glass fiber face-sheet	4	(0°, 90°, 90°, 0°)	40.742	1	23.872
Epoxy woven carbon fiber face-sheet	2	(0°, 90°)	27.069	0.6	24.272
Hybrid composite face-sheets	4	(0°, 90°, 90°, 0°)	40.115	1.1	23.772

**Table 12 polymers-13-00834-t012:** Minimum weight objective function with optimum face-sheet thickness and core thickness for the sandwich base plate of military aircraft pallets consists of an aluminum honeycomb core and orthotropic FRP composite face-sheets [(1) Phenolic woven glass fiber, (2) Epoxy woven glass fiber, (3) Epoxy woven carbon fiber and (4) Hybrid layers] with different number of layer $N_{l}$ and fiber orientation $\theta^{{}^{\circ}}$.

Type	Number of Layers *N_l_*	Fiber Orientations θ°	Wmin,t [kg]	tf,opt [mm]	tc,opt [mm]
	Optimum Value				
Phenolic woven glass fiber face-sheet	4	(0°, 90°, 90°, 0°)	40.742	1	23.872
Epoxy woven glass fiber face-sheet	4	(0°, 90°, 90°, 0°)	40.742	1	23.872
Epoxy woven carbon fiber face-sheet	2	(0°, 90°)	27.069	0.6	24.272
Hybrid composite face-sheets	4	(0°, 90°, 90°, 0°)	40.115	1.1	23.772

**Table 13 polymers-13-00834-t013:** Annual fuel, carbon savings and total savings for the FRP composite sandwich base plate compared to the conventional aluminum base plate of aircraft pallets.

1. Fuel Savings	Price	Unit
Weight of fuel required to carry 1 kg additional weight per hour	0.04	kg
Expected annual hours flown	5000	h
Weight of fuel required to carry 1 kg weight for one year	200	kg
Current cost of fuel per 1000 kg (from Jet fuel price monitor)	812	€
Annual cost to carry 1 kg additional weight for one year	162	€
Quantity of units per aircraft	26	unit
Quantity of shipsets	4	set
Weight of conventional aluminum pallet	80	kg
Number of units required	104	unit
Weight of light-weight sandwich FRP pallet (optimal result)	27	kg
Weight reduction in one pallet	53	kg
Fuel cost saving per year for one pallet	8586	€
Weight reduction in one aircraft	1378	kg
Fuel cost saving per year for one aircraft	223,787	€
**2. Carbon Savings**	**Price**	**Unit**
Carbon produced per kg of fuel	3.1	kg
Total carbon produced to carry 1 kg for one year	620	kg
Total carbon saving	854,360	kg
Cost of carbon per Ton	32.71	€
Annual carbon cost saved	27,943	€
**3. Total Savings**	**Price**	**Unit**
Combined effect of reduced fuel consumption and carbon reduction	251,730	€

**Table 14 polymers-13-00834-t014:** Safety factors for optimum design constrains for the single base plate of military aircraft pallet.

Constraints		Factor of Safety (FoS)		
		Epoxy Woven Glass Fiber Face-Sheet 4-Layers (0°, 90°, 90°, 0°)	Epoxy Woven Carbon Fiber Face-Sheet 2-Layer (0°, 90°)	Hybrid Composite Face-Sheet 4-Layers (0°, 90°, 90°, 0°)
Bending stiffness	$D_{11,x}$	4.92	10.11	12.32
Total deflection	δ	4.86	9.84	11.93
Skin stress (bending load)	$\sigma_{f}$	2.23	1.89	2.29
Core shear stress	$\tau_{c}$	1	1	1
Facing stress (end loading)	$\sigma_{f}$	13.68	11.05	13.55
Overall buckling	$P_{b,\;cr}$	1.74	3.52	4.27
Shear crimping	$P_{cr}$	70.39	71.56	70.09
Skin wrinkling critical stress in x-directions	$\sigma_{wr,\;cr}$	1.81	1.94	2.37
Skin wrinkling critical stress in y-directions	$\sigma_{wr,\;cr}$	1.56	1.76	2.14
Skin wrinkling critical load	$P_{wr,cr}$	9.99	8.33	17.28
Intracell buckling	$\sigma_{f,\;cr}$	1.87	1.65	4.47

## Data Availability

The data presented in this study are available on request from the corresponding author.

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
