# Peer review of "Optimal Design of a Fiber-Reinforced Plastic Composite Sandwich Structure for the Base Plate of Aircraft Pallets In Order to Reduce Weight"

_polymers, 2021, doi:10.3390/polym13050834_

Round 1

Reviewer 1 Report

Overall, my review is positive and includes the following aspects to be appreciated:

  • the introduction provide sufficient background in area of Fiber Reinforced Plastic Composite Sandwich Structures;
  • the paper contain results on experimental tests and numerical calculations for the investigated FRP sandwich panels, results in weight savings of base plates of aircraft pallets. Also, these results validate the applicability of the used calculation method;
  • the mechanical properties of 40 different layer-combinations of 4 different FRP face-sheet materials (phenolic woven glass fiber, epoxy woven glass fiber, epoxy woven carbon fiber and hybrid layers) were investigated using the Digimat-HC modeling program in order to find the appropriate face-sheet construction;
  • the validation of the optimization procedure carried out using Excel Solver (Generalized Reduced Gradient Nonlinear Algorithm) and the Matlab (Interior Point Algorithm) programs; the efficiency of the newly elaborated optimization method was confirmed by the case study;
  • the “Materials and Methods” section is well-structured; also, the “Numerical Results” and “Graphical Presentation” sections are well-presented, offering various comparisons;
  • the conclusions are supported by the experimental results.

Reviewer 2 Report

Dear Editor.

The authors investigated the weight saving if we replace the aluminum materials with the fiber reinforced plastics. They showed the case study for the optimal FRP type, and it should be of industrial interest. However, I do not recommend it for publication in Polymers, because there are critical problems as listed below.

  1. The authors chose four different materials: phenolic woven glass fiber, epoxy woven glass fiber, epoxy woven carbon fiber, and hybrid layers. However, the difference among these materials is not explained sufficiently. Thus, I cannot judge the validity of the difference in the simulation results. The authors should show the bulk properties such as Young’s modulus, stress-strain curves, glass transition etc. If not, the readers cannot understand whether the difference is originating from higher-order structure or material toughness.

  1. In the section 2, although the title is materials and method, many results and the explanation of the previous studies were shown, which is not appropriate. The authors should reconsider the structure of the article.

  1. They showed the data with too many significant digits in Tables. The author should show the reliability of these data in the experimental section.

Reviewer 3 Report

Introduction:

  • "Epoxy resins is....with formaldehyde", this section is not so relevant with the study.
  • "The main reason for the....traditional metals", you have already highlighted this before, do not repeat

Materials and Methods:

  • 2nd paragraph is more suited in Introduction section
  • Tables 5,6 are identical, it seems wrong to me

Please clarify how many sandwich panels have you studied, 3 or 4? In Figure 6 , you have plotted three curves. Optimization methods are not novel, please correct this. Regarding, design and optimisation, you should have optimised versus fibres orientation as well.  Safety factor values after optimisation are not reported. You should comment on this.

Reviewer 4 Report

This research investigates optimal Design for FRP Composite Sandwich Structures utilized in the base plate of aircraft pallets. The weight reduction has been considered in the analysis. A case study for the base plate of an aircraft pallet was introduced for the validation of the optimization procedure carried out using the Matlab (Interior Point Algorithm) and Excel Solver (Generalized Reduced Gradient Nonlinear Algorithm) programs.  

This is an interesting work due to its engineering application. However, in my opinion a number of important issues should still be addressed through a major revision, in the interest of improving the present manuscript, as follows:

(1) The authors haven't even mentioned anything about their experimental investigation much less discussing the experimental results in their abstract. Thus, a crucial revision of the text should be considered.  

(2) It is not clear how the numerical simulation has been conducted. Many information is missing. Since the authors have presented the simplified geometry model of the prop after meshing, I assume that the finite element (FE) simulation has been adapted. If not it must be clearly mentioned.  

(3) The authors haven't provided detailed analysis as to how the numerical simulation analysis was conducted. It's extremely important. The authors are encouraged to discuss how the FE simulation analysis was conducted. Some issues to be considered:   -How were the loading and composite material properties introduced into the FE software? -How were the boundary conditions prescribed to your FE model?  -What type of element has been selected from the material library? exp: shell elements if plane stress is considered for thin plates or solid elements for thick and/or moderately thick plates. -Have you done mesh sensitivity analysis to warrant the results convergence and accuracy?  

(4) What is the point of having experimental investigation when there is no proper results comparison? Please clarify.  

(5) The entire experimental apparatus and test procedures must be provided. 

(6) The Introduction doesn't provide useful information about the core objective of this research when it comes to the numerical (FE) simulations of composite materials and structures. It appears that the authors aren't much familiar with some research publications associated with various engineering structures under various loading conditions.  I have provided two useful references for the author's attention for their introduction section.   "Localized failure analysis of internally pressurized laminated ellipsoidal woven GFRP composite domes: Analytical, numerical, and experimental studies", Archives of Civil and Mechanical Engineering, Vol:19, pp:1235-1250.     "Finite element analysis of grinding process of long fiber reinforced ceramic matrix woven composites: Modeling, experimental verification and material removal mechanism, Ceramics International Vol: 45, pp: 15920-15927.  

(7) What is the use of Figure 9. Base plate of conventional aluminum sheet aircraft pallet? The reviewer believes it doesn't contribute to this research and hence must be removed.  

(8) The manuscript requires significant language revision. There are numerous typos and wrong sentence structures throughout the manuscript. Authors can benefit from having their paper revised and edited by a native English speaker to improve its language quality.       

Round 2

Reviewer 2 Report

Thank you for giving me another chance to review this paper. The authors revised the manuscript. I validated their revision. And I understood that they are interested in the macroscopic structure of "composite materials" and not interested in the polymer characteristics.